# Antinociceptive Effects of Cannabichromene (CBC) in Mice: Insights from von Frey, Tail-Flick, Formalin, and Acetone Tests

**DOI:** 10.3390/biomedicines12010083

**Published:** 2023-12-29

**Authors:** Wesley M. Raup-Konsavage, Diana E. Sepulveda, Jian Wang, Nikolay V. Dokholyan, Kent E. Vrana, Nicholas M. Graziane

**Affiliations:** 1Department of Pharmacology, Penn State College of Medicine, Hershey, PA 17033, USA; 2Department of Anesthesiology & Perioperative Medicine, Penn State College of Medicine, Hershey, PA 17033, USA; 3Department of Biochemistry & Molecular Biology, Penn State College of Medicine, Hershey, PA 17033, USA; 4Department of Chemistry, Penn State University, University Park, PA 16802, USA; 5Department of Biomedical Engineering, Penn State University, University Park, PA 16802, USA

**Keywords:** neuropathic pain, cannabichromene, pain, inflammatory pain, cannabinoids, cannabis

## Abstract

*Cannabis sativa* contains minor cannabinoids that have potential therapeutic value in pain management. However, detailed experimental evidence for the antinociceptive effects of many of these minor cannabinoids remains lacking. Here, we employed artificial intelligence (AI) to perform compound–protein interaction estimates with cannabichromene (CBC) and receptors involved in nociceptive signaling. Based on our findings, we investigated the antinociceptive properties of CBC in naïve or neuropathic C57BL/6 male and female mice using von Frey (mechanical allodynia), tail-flick (noxious radiant heat), formalin (acute and persistent inflammatory pain), and acetone (cold thermal) tests. For von Frey assessments, CBC dose (0–20 mg/kg, i.p.) and time (0–6 h) responses were measured in male and female neuropathic mice. For tail-flick, formalin, and acetone assays, CBC (20 mg/kg, i.p.) was administered to naïve male and female mice 1 h prior to testing. The results show that CBC (10 and 20 mg/kg, i.p.) significantly reduced mechanical allodynia in neuropathic male and female mice 1–2 h after treatment. Additionally, CBC treatment caused significant reductions in nociceptive behaviors in the tail-flick assay and in both phase 1 and phase 2 of the formalin test. Finally, we found a significant interaction in neuropathic male mice in the acetone test. In conclusion, our results suggest that CBC targets receptors involved in nociceptive signaling and imparts antinociceptive properties that may benefit males and females afflicted with diverse forms of acute or chronic/persistent pain.

## 1. Introduction

The management of pain remains a major health concern globally, and due in part to the opioid epidemic, there is a growing interest in the development of novel compounds that can be used to treat and manage pain [1,2,3,4]. Research has been conducted on the use of cannabis or individual cannabinoids to treat pain, including Δ^9^-tetrahydrocannabinol (THC), cannabidiol (CBD), cannabigerol (CBG), and cannabinol (CBN) [5,6,7,8,9,10,11,12,13,14,15,16,17,18,19]. In addition to these cannabinoids, cannabichromene (CBC) is a non-psychoactive minor cannabinoid (cannabinoid compounds found in the cannabis plant at low concentrations) that has been reported to have anti-microbial and anti-inflammatory activities [20,21,22,23,24,25,26].

CBC is one of three major cannabinoids enzymatically produced from CBG by cannabis plants, along with THC and CBD [27,28]. CBC has been reported to act as an agonist of the CB2 receptor, as well as the TRPA1, TRPV1, TRPV3 and TRPV4 ion channels [29,30,31,32]. Additionally, CBC binds to CB1 receptors and peroxisome proliferator-activated receptors (PPARs) [33,34,35]. It also acts as a low-potency antagonist at TPRM8 channels [29,30]. Several of these receptors play a role in reducing pain and are the targets of current therapeutic agents, further suggesting that CBC may be a novel antinociceptive agent. Despite this, current knowledge on the antinociceptive properties of CBC remains limited, with only two studies suggesting CBC can reduce thermal pain [21,23].

To begin our assessment of CBC as an antinociceptive agent, we employed artificial intelligence (AI) software, Drug-Target Identification (DRIFT) [36,37], to estimate compound–protein interactions between CBC and nociceptive receptors. We compared CBC’s interaction profile with cannabinoids possessing antinociceptive properties, including cannabinol (CBN), Δ^9^-tetrahydrocannabinol (THC), cannabigerol (CBG), cannabidiol (CBD), and Δ^9^-tetrahydrocannabivarin (THCV). Subsequently, we conducted a series of behavioral assays in mice to assess the ability of CBC to act as an antinociceptive agent. Our findings reveal that CBC reduces behaviors associated with pain across a wide range of noxious stimuli, including inflammatory pain, radiant heat-induced pain, and neuropathic pain.

## 2. Materials and Methods

### 2.1. DRIFT Analysis

The phytocannabinoids cannabinol (CBN), Δ^9^-tetrahydrocannabinol (THC), cannabigerol (CBG), cannabidiol (CBD), Δ^9^-tetrahydrocannabivarin (THCV), and cannabichromene (CBC) were evaluated by DRIFT. This artificial intelligence/machine learning tool (AI/ML) is designed to predict potential protein targets of individual small molecules, as previously described [36,37].

### 2.2. Animals

The total number of mice used in this study included male (*N* = 105) and female (*N* = 106) age-matched (10–12 weeks) C57BL/6 wild-type mice (The Jackson Laboratory, Bar Harbor, ME, USA). All mice were group-housed on a 12 h light/dark cycle with ad libitum food and water. All experimental procedures were approved by the Pennsylvania State University College of Medicine Institutional Animal Care & Use Committee (IACUC).

### 2.3. Drugs

Formalin solution was prepared from 37% formaldehyde stock solution (Thermo Fisher Scientific, Cat #: F79; Waltham, MA, USA). Indomethacin was purchased from Alfa Aesar (Haverhill, MA, USA), cisplatin solution was purchased from Acros Organics (Fairlawn, NJ, USA), and CBC was purchased from Cayman Chemical (Ann Arbor, MI, USA).

### 2.4. Cisplatin-Induced Peripheral Neuropathy

Cisplatin-induced peripheral neuropathy was evoked using previously described methods [17,38]. Male (*n* = 24) and female (*n* = 24) mice were injected with cisplatin (5 mg/kg, i.p.) once a week for 4 weeks. Immediately after cisplatin injection, mice were administered 4% sodium bicarbonate solution (1 mL, s.c.) to prevent nephrotoxicity and to minimize damage to renal function [38]. Mechanical allodynia was assessed before and after cisplatin treatment to confirm neuropathic pain state.

### 2.5. von Frey Assay

Mechanical hypersensitivity was assessed using an electronic von Frey anesthesiometer (IITC Life Sciences Inc., Woodland Hills, CA, USA) as previously described [17]. Three tests were performed and averaged, and each test was separated by at least 3 min. To measure the effects of test compounds, neuropathic mice were randomly assigned to groups and injected with vehicle (DMSO, Tween 80, saline [1:1:18], i.p.), CBC (5, 10, or 20 mg/kg, i.p.), or indomethacin (10 mg/kg, i.p.) 1 h prior to tests. Indomethacin was used as a positive control based on our studies and those from others showing that indomethacin significantly reduces nociception in murine models of neuropathic pain [17,39,40]. Experimental raters were all blinded to the treatment condition during assessments.

### 2.6. Formalin Test

Formalin tests were performed as previously described [17]. Mice received injections of vehicle (DMSO, Tween 80, saline (1:1:18), i.p.), CBC (20 mg/kg, i.p.), or indomethacin (20 mg/kg, i.p.) 1 h prior to tests by an experimenter blinded to treatment. Male (*n* = 19) and female (*n* = 20) mice were randomly assigned to groups. Mice were acclimated to a plexiglass observation chamber (in cm: 13 × 13 × 13) on a transparent table for 20 min. After the 20 min acclimation period, 2.5% formalin solution (10 μL) was injected to the plantar surface of the right hind paw. Mice were immediately placed back into the plexiglass observation chamber. Mouse behavior was recorded for 60 min using a GoPro camera (San Mateo, CA, USA), which was placed underneath the observation chamber. Four behavioral categories were observed by a trained, blinded observer: (i) no behavior, (ii) little or no weight placed on the injected paw, (iii) raised injected paw, and (iv) licking, shaking, biting, or rapid lifting of the injected paw. These behaviors were assessed during twelve 5 min (i.e., 300 s) bins. The time that the animal spent displaying each behavior was recorded and a weighted composite score was calculated based on the following formula for each 5 min bin, as previously published [11,15,41]: ((a + b) + (2 × c))d, where “a” refers to the time the animal spent placing little or no weight on the injected paw, “b” refers to the time the animal spent raising the injected paw, “c” refers to the time the animal spent licking, shaking, biting or rapid lifting of the injected paw, and “d” refers to the total time (i.e., 300 s). The weighted composite score was selected based on the severity of pain associated with “c.” Based on this formula, the maximum composite score would equal 2, signifying a maximum pain state. The area under the curve (AUC) was calculated for the acute phase (phase 1) (0–15 min) and the inflammatory phase (phase 2) (15–60 min) using GraphPad Prism (9.5.1).

### 2.7. Tail-Flick Assay

Thermal hyperalgesia was measured using a Columbus Instruments TF-1 tail-flick analgesia meter (Columbus, OH, USA), using previously described methods [11,17,18]. Male (*n* = 15) and female (*n* = 15) mice were acclimated to the apparatus 1 day prior to testing to reduce stress. Acclimation and baseline testing measurements were repeated until the time of tail withdrawal fell within 0.5 s of the previous measurement (e.g., 3.20, 3.57 s) or until 5 measurements were taken. After baseline measurements, mice received injections of vehicle (DMSO, Tween 80, saline (1:1:18), i.p.), CBC (20 mg/kg, i.p.), or indomethacin (10 mg/kg, i.p.)) 1 h prior to tests by an experimenter blinded to treatment. Mice were randomly assigned to groups. The tail-flick latency was measured by calculating the percent maximal possible effect (%MPE) (%MPE = (post-drug latency–pre-drug latency)/(10-pre-drug latency)).

### 2.8. Acetone Test

Cold thermal pain was assessed by placing male and female mice (*n* = 8 for each) in a plexiglass observation chamber (13 × 13 × 13 cm) on a mesh table (IITC Life Sciences Inc., Woodland Hills, CA, USA). Acetone (10 μL) was placed on the right hind paw of the mouse and the time (in seconds) of a behavioral response (licking, biting, shaking the hind paw) was measured. This process was repeated at each time point (30 min, 1 h, 2, 4 and 6 h post-injection). Mice were injected with either vehicle (DMSO, Tween 80, saline (1:1:18), i.p.) or CBC (20 mg/kg, i.p.).

### 2.9. Statistical Analyses

All results are shown as means ± SEM. No data points were excluded. Statistical significance was assessed in GraphPad Prism software (9.5.1) using Student’s *t*-test, one-way analysis of variance (ANOVA), two-way ANOVA, or two-way repeated-measures ANOVA with Bonferroni correction for multiple comparisons to identify differences as specified. *F*-values for two-way ANOVA statistical comparisons represent interactions between variables. Two-tailed tests were performed for all studies.

## 3. Results

### 3.1. CBC’s Receptor Binding Profile in Comparison to Other Phytocannabinoids

Using DRIFT software (version 1) [36], we evaluated a number of phytocannabinoids for potential receptor targets that might be involved in treating pain. We quantified the similarity between CBN, THC, CBG, CBD, THCV, and CBC by computing the Pearson correlation coefficients based on their respective DRIFT score profiles across a list of target proteins (Figure 1A). A large coefficient value (above 0.5) suggests a strong similarity in the binding profiles of the two compounds, indicating that they may interact with similar receptor targets. Conversely, a lower coefficient value (0.1–0.3) implies a weak to low correlation. We observed that CBC has a significant positive correlation with other phytocannabinoids examined, including CBN (r = 0.2679; *p* < 0.0001), CBG (r = 0.2539; *p* < 0.0001), CBD (r = 0.1557; *p* = 0.0009), THC (r = 0.3102; *p* < 0.0001), and THCV (r = 0.3160; *p* < 0.0001). However, the strength of the correlation is low, as shown by the Pearson’s correlation coefficient (r), suggesting a weak similarity in its receptor binding profile compared to other phytocannabinoids.

Evidence suggests that THC, CBD, and CBG are effective antinociceptive agents with binding affinities for many receptors involved in pain modulation [5,6,7,8,9,10,11,12,13,15,17,18,42]. Given that CBC is derived from the same plant and possesses a significant positive correlation in its binding profile compared to other phytocannabinoids (Figure 1A), we next investigated whether CBC had binding affinity for receptors known to play a role in pain modulation. DRIFT software identified binding and docking sites between CBC and many receptors known to have a role in generating or mitigating pain, including TRPA1, CB1, CB2, and Cox1 (Figure 1B). Therefore, we examined the analgesic properties of CBC by conducting pain assays in mice.

### 3.2. CBC Reduces Mechanical Sensitivity in Neuropathic Male and Female Mice

In order to investigate the antinociceptive efficacy of CBC, we measured mechanical sensitivity using the von Frey test in neuropathic male and female mice. For these experiments, mechanical sensitivity was assessed prior to and following cisplatin injections (baseline and post-cisplatin, respectively) (Figure 2A). Approximately three days after the last cisplatin injection, mice received vehicle, CBC at varying doses (5, 10, or 20 mg/kg, i.p.), or the positive control indomethacin (10 mg/kg, i.p.) 1 h prior to the von Frey test (Figure 2A). The results show that in comparison to the vehicle control, CBC at 10 and 20 mg/kg caused a significant increase in the force required to evoke a paw withdrawal response in both neuropathic male (F_(4,51)_ = 23.76, *p* < 0.0001; one-way ANOVA with Bonferroni post-test) (Figure 2B) and female mice (F_(4,51)_ = 26.70, *p* < 0.0001; one-way ANOVA with Bonferroni post-test) (Figure 2C). Additionally, the results show that at 20 mg/kg, CBC produced a greater effect in neuropathic male mice when compared to females (F_(3,60)_ = 73.89, *p* < 0.0001; one-way ANOVA with Bonferroni post-test) (Figure 2D).

We next investigated the time course of CBC actions by performing the von Frey test on neuropathic male and female mice at varying time points (30 min, 1 h, 2 h, 4 h, and 6 h) following CBC injection (20 mg/kg, i.p.). We found no significant differences between vehicle control or CBC groups prior to (baseline) (*p* > 0.9999; Bonferroni post-test) or following cisplatin treatment (post-cisp.) (*p* > 0.9999; Bonferroni post-test) in either neuropathic male (Figure 2E) or female mice (Figure 2F). These results suggest that there were no differences between groups prior to treatment. Additionally, the results reveal that CBC (20 mg/kg, i.p.) produced a significant increase in the force required to evoke a paw withdrawal response 1 h following CBC treatment when compared to vehicle control in both neuropathic male (*p* < 0.001; Bonferroni post-test) (Figure 2E) and female mice (*p* < 0.0001; Bonferroni post-test) (Figure 2F). This significant reduction did not completely reverse the mechanical allodynia in male mice, as we observed a significant difference between CBC (20 mg/kg) compared to pre-cisplatin levels (baseline) (*p* = 0.0018; Bonferroni post-test) (Figure 2E). However, in females, CBC at 1 h post-treatment reversed mechanical allodynia, as there was no significant difference in the force required to evoke a paw withdrawal response when compared to baseline (*p* > 0.9999; Bonferroni post-test) (Figure 2F). No significant differences were observed at the other time points.

### 3.3. CBC Produces Analgesia-like Effects in Male and Female Mice: Formalin Test (Acute and Inflammatory Pain)

The discovery that CBC demonstrated efficacy in reducing mechanical hypersensitivity in neuropathic mice led us to explore whether CBG would exhibit effectiveness in mitigating behaviors in alternative pain models. We next explored the antinociceptive effects of CBC in naïve male and female C57BL/6 mice undergoing the formalin test. Mice were treated with vehicle or CBC (20 mg/kg, i.p.) 1 h prior to formalin injection. Our results show that formalin produced two clear phases of pain, with phase 1 occurring from 0–15 min after formalin injection and phase 2 occurring from 15–60 min after formalin injection (Figure 3A and Figure 4A). These results are consistent with previous findings [15,43]. When analyzing the area under the curve (AUC) for both phase 1 and phase 2 of the formalin test, we found that CBC significantly reduced the AUC in both phases in male (Figure 3B,C) and female (Figure 4B,C) mice.

### 3.4. CBC Produces Analgesia-like Effects in Male and Female Mice: Tail-Flick Test (Noxious Radiant Heat)

We next examined the analgesic effects of CBC on a thermal stimulus. Naïve C57BL/6 mice underwent the tail-flick assay 1 h following injection with vehicle, CBC (20 mg/kg, i.p.), or the positive control, indomethacin (10 mg/kg, i.p.) (Figure 5A). Our results show that CBC significantly increased the percentage maximum possible effect (%MPE) compared to vehicle control in both male (*p* < 0.0001; Bonferroni post-test) (Figure 5B) and female mice (*p* = 0.0008; Bonferroni post-test) (Figure 5C). Additionally, we found that CBC’s analgesic effects in male and female mice were similar to indomethacin, as no significant differences between groups were observed (males: *p* = 0.1863; females: *p* > 0.9999; Bonferroni post-test) (Figure 5B,C).

### 3.5. CBC Effects in Male and Female Mice: Acetone Test (Cold Thermal Pain)

Finally, we investigated the analgesia-like effects of CBC on naïve male and female C57BL/6 mice undergoing the acetone test (cold sensitivity). For these experiments, mice were treated with vehicle, CBC (20 mg/kg, i.p.), or the positive control, indomethacin (10 mg/kg, i.p.), 1 h prior to exposure to acetone placed on the hind paw. The time required for a response was measured. The results show that unlike indomethacin, CBC had no statistically significant effect on the time for response in either male or female mice (Figure 6A,B). Given the well-established role of pain in sensory perception and behavioral responses, we hypothesized that the absence of a pain state in naïve mice might have contributed to the lack of response. Therefore, we conducted experiments in a separate group of mice that expressed neuropathy. Here, following 4 weeks of cisplatin injections, male and female mice were treated with vehicle or CBC (20 mg/kg, i.p.) and the time of response after acetone application was measured at varying time points (30 min, 1 h, 2 h, 4 h, and 6 h) (Figure 6C,D). We found a significant interaction in neuropathic male mice (F_(4,56)_ = 2.596, *p* = 0.0459; two-way repeated-measures ANOVA with Bonferroni post-test) (Figure 6C). Post hoc analysis in neuropathic male mice revealed no significant differences between groups (vehicle vs. CBC) at any time points (Figure 6C). Unlike in neuropathic male mice, no significant differences were observed in neuropathic female mice (F_(4,56)_ = 1.974, *p* = 0.1110; two-way repeated-measures ANOVA with Bonferroni post-test) (Figure 6D).

## 4. Discussion

Here, we found that CBC was effective at reducing pain in male and female neuropathic mice, as well as reducing acute, inflammatory, and radiant heat pain in male and female naïve animals. This study demonstrates the broad range of the antinociceptive activities of CBC.

Interestingly, the antinociceptive properties of CBC occurred in neuropathic mice undergoing von Frey tests, in the formalin assay (acute and inflammatory), and in the tail-flick assay (noxious radiant heat). This suggests that CBC may possess broad pain-relieving properties, which are unique among minor cannabinoids. For example, previously, we found that another minor cannabinoid, CBG, was effective at reducing neuropathic pain in male and female mice, but less effective at reducing pain in the formalin and tail-flick assays [17,42]. Additionally, in a previous study, we found that CBD’s antinociceptive effect occurred only during the acute phase of the formalin assay in male mice (not in female mice) and that CBD was not effective at reducing mechanical hypersensitivity in a mouse model of chemotherapy-induced peripheral neuropathy (CIPN) [15,18]. Despite these differences from other minor cannabinoids, the observed antinociceptive effects of CBC are consistent with the wide range of antinociceptive properties associated with THC treatment [9,12,16,44,45]. However, unlike THC, CBC at the dose used in this study does not induce catalepsy in mice [12,18,46]. The absence of catalepsy at low CBC doses implies that CBC may be a more suitable cannabinoid for pain reduction than cannabis, known for its elevated levels of THC.

The predictions from DRIFT identified potential receptor targets for CBC, including CB1 receptors, CB2 receptors, and TRPA1 channels, which align with empirical research [23,29,30,32,35,46]. Furthermore, the absence of TRPM8 on the list generated by DRIFT aligns with empirical studies evaluating phytocannabinoid activity in TRP channels. In these studies, CBC exhibited an IC_50_ of over 40 µM at TRPM8 [30]. The low potency of CBC for TRPM8 is further supported by the lack of clear CBC effect in the cold thermal assay observed in this study. For example, we did not observe a significant effect of CBC in the acetone test in naïve mice and only observed a statistical interaction in neuropathic male mice, likely due to transient responses that occurred within the first 30 min post-CBC treatment.

Along these lines, it is interesting to observe the effects of CBC over time in each behavioral assay. Whereas a weak activity with receptors/channels involved in cold thermal pain may lead to transient antinociceptive effects, stronger activation of receptors involved in mechanical hypersensitivity, inflammation, and/or heat pain may result in prolonged antinociceptive effects. For example, von Frey tests revealed a maximum antinociceptive effect of CBC at 1 h post-treatment with no effect at 30 min post-treatment in neuropathic mice. This finding is comparable to the time courses of other minor cannabinoids, including CBD and CBG [17,18], suggesting CBC has a similar pharmacokinetic profile [47]. Additionally, it suggests that CBC activates receptors/channels involved in mechanical hypersensitivity, potentially including TRPA1 receptors and CB2 receptors [48,49,50,51,52,53,54,55,56].

In the formalin assay, CBC was administered 1 h prior to formalin injection and the effects of CBC were observed throughout the duration of the 60 min. period. This 60 min period consists of an acute phase mediated by C-fiber activation followed by a prolonged phase mediated by inflammatory factors, including substance P, bradykinin, histamine, serotonin, and prostaglandins [43,57]. These inflammatory factors activate TRPA1 channels directly or indirectly through signaling pathways, leading to the nociceptive behaviors in the formalin assay [58,59]. Given that CBC is a potent agonist and desensitizer of TRPA1 channels [30], it is plausible that our observed CBC-induced antinociceptive effect during phase 2 of the formalin assay is mediated by prolonged desensitization of TRPA1 channels, thus silencing them in the presence of inflammatory factors. Additionally, CB2 receptors are expressed on immune cells (B cells, natural killer cells, monocytes, neutrophils, CD8 lymphocytes, and CD4 lymphocytes), inhibit proinflammatory cytokine production, and increase anti-inflammatory cytokines [60,61,62]. Therefore, the robust antinociceptive effects exhibited by CBC during the second phase of the formalin assay may also be attributed to its agonistic action on CB2 receptors [32,63].

Similarly, in the tail-flick assay, CBC was administered 1 h prior to tests and produced a significant antinociceptive effect in both males and females, which was comparable to the effects observed by the positive control indomethacin. Evidence suggests that the response to noxious heat employed during the tail-flick assay is mediated by a reflex organized at the level of the spinal cord [64] and activates type II Aδ myelinated fibers that project to lamina I and V of the spinal cord [59]. This heat activated pain response is mediated, in part, by TRPV channels. Empirical evidence suggests that CBC is an agonist at TRPV3 and TRPV4 channels [65], and it may be through CBC-induced desensitization of TRPV3/4 channels that mediates the antinociceptive effects in the tail-flick assay. Alternatively, based on the antinociceptive effects observed following treatment with indomethacin, CBC may be acting through anti-inflammatory mechanisms. In addition to the receptor targets for CBC identified by DRIFT, DRIFT also identified potential novel CBC targets, such as cyclooxygenase 1 (COX-1), which—when inhibited by nonsteroidal anti-inflammatory drugs (NSAIDs)—reduce prostaglandin production, resulting in anti-inflammation. Evidence suggests that prostaglandins sensitize nociceptors involved in noxious heat detection [66]. Therefore, CBC’s effects during the tail-flick assay may be mediated through modulation of nociceptive fibers in the periphery, which would be expected to influence a reflex mediated at the level of the spinal cord.

Lastly, our results in combination with other published findings demonstrate that the broad range of CBC actions in antinociception are potentially mediated by CBC’s ability to interact with multiple receptors/channels, including CB2 receptors, TRPA1 channels, and TRPV channels. However, CBC has also been reported to activate PPAR receptors, and the activation of these receptors has been found to reduce neuropathic pain in animal models [33,67,68,69]. Additionally, the activation of PPAR receptors has been found to reduce inflammation and potentially inflammatory pain [70,71]. Because CBC has been reported to be an agonist of PPARγ, this pathway may also be involved in mediating the antinociceptive properties of CBC observed in many of the assays performed in this study. Overall, integrating the results from AI that identified CBC receptor targets with the outcomes observed in a variety of pain assays may guide future experiments aimed at elucidating the receptors involved in CBC antinociceptive actions.

## 5. Conclusions

In the current animal study, we set out to evaluate the antinociceptive properties of CBC. Our results show that CBC is effective at reducing pain throughout many pain assays in both male and female mice, revealing CBC as a potential therapeutic option for treating not only thermal pain but also other diverse types of pain (neuropathic and inflammatory pain). Future studies to identify receptors that mediate the antinociceptive effects of CBC will be beneficial in understanding the mechanism by which CBC acts.

## Figures and Tables

**Figure 1 biomedicines-12-00083-f001:**
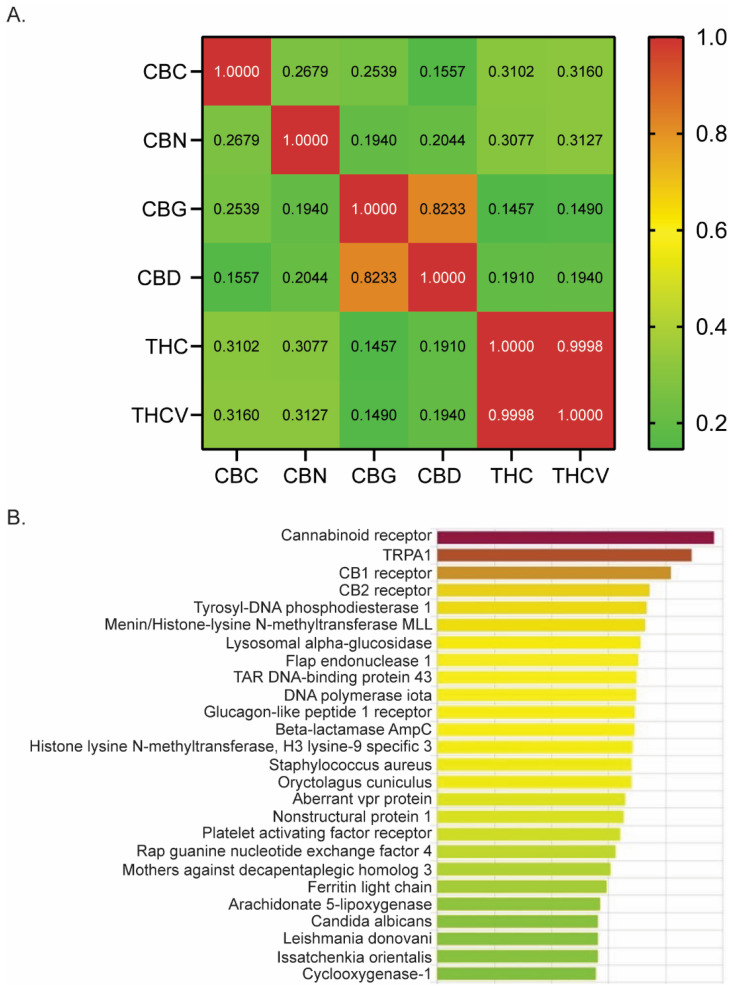
DRIFT predicts CBC’s receptor binding profile. (**A**) Pearson correlation analysis of six phytocannabinoids examined by DRIFT. (**B**) DRIFT prediction of CBC targets. Compounds are ranked from highest (red) to lowest (green) confidence of prediction. CBN (cannabinol), THC (Δ^9^-tetrahydrocannabinol), CBG (cannabigerol), CBD (cannabidiol), THCV (Δ^9^-tetrahydrocannabivarin), CBC (cannabichromene).

**Figure 2 biomedicines-12-00083-f002:**
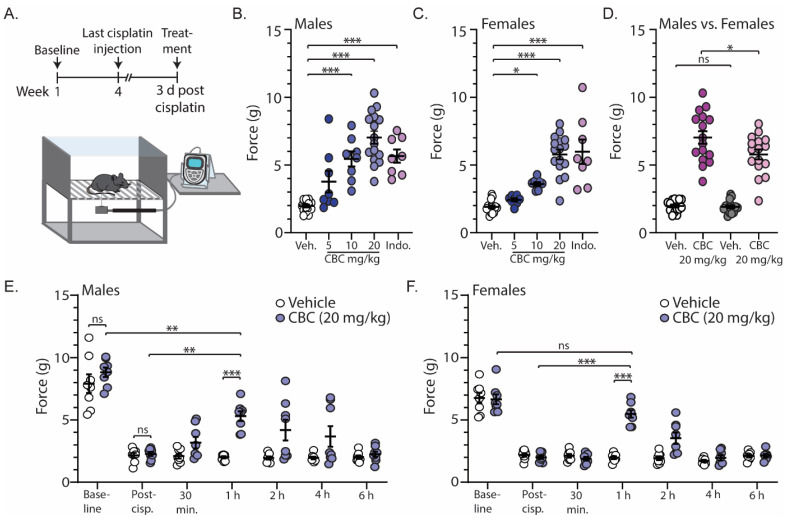
CBC reduces mechanical sensitivity in neuropathic male and female mice. (**A**) Timeline of experimental procedure for von Frey experiments. (**B**) Summary graph showing the force required to evoke a paw withdraw response in neuropathic male mice 1 h after treatment with vehicle, varying doses of CBC (5, 10, 20 mg/kg, i.p.), or indomethacin (10 mg/kg, i.p.) (F_(4,51)_ = 23.76, *p* < 0.0001; one-way ANOVA with Bonferroni post-test). (**C**) Summary graph as described in (B) in neuropathic female mice (F_(4,51)_ = 26.70, *p* < 0.0001; one-way ANOVA with Bonferroni post-test). (**D**) Summary graph comparing the effects of 20 mg/kg CBC in neuropathic male and female mice (F_(3,60)_ = 73.89, *p* < 0.0001; one-way ANOVA with Bonferroni post-test). (**E**) Summary graph showing the force required to evoke a paw withdraw response in neuropathic male mice before cisplatin treatment (baseline), after cisplatin treatment (post-cisp.), and at varying time points after treatment with vehicle or CBC (20 mg/kg, i.p.) (F_(6,84)_ = 3.797, *p* = 0.0021; two-way repeated-measures ANOVA with Bonferroni post-test). (**F**) Summary graph described in (**E**) in neuropathic female mice (F_(6,84)_ = 15.85, *p* < 0.0001; two-way repeated-measures ANOVA with Bonferroni post-test). ns = not significant, * *p* < 0.05, ** *p* < 0.01, *** *p* < 0.001.

**Figure 3 biomedicines-12-00083-f003:**
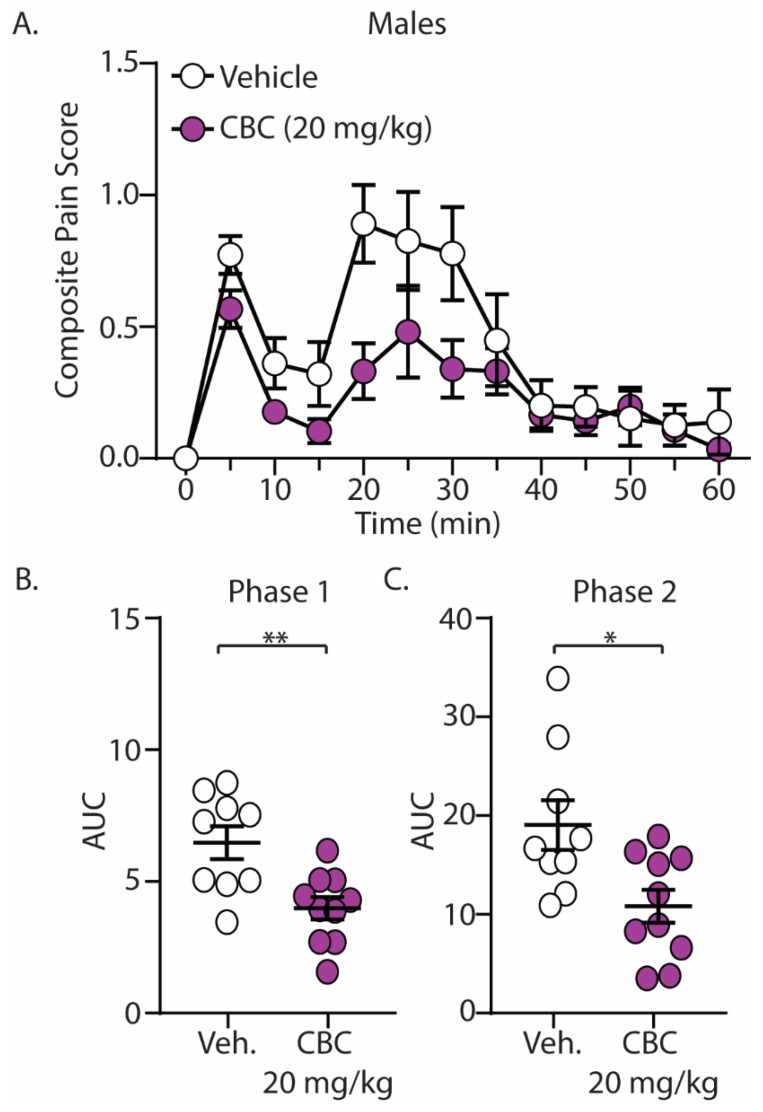
CBC produces analgesia-like effects in male mice in the formalin test. (**A**) Summary of the composite pain score 0–60 min after formalin injection in male mice treated with vehicle or CBC (20 mg/kg, i.p.) (F_(12,204)_ = 1.731, *p* = 0.0624; two-way repeated-measures ANOVA with Bonferroni post-test). (**B**) Summary graph showing the area under the curve (AUC) during phase 1 of the formalin assay in male mice treated with vehicle or CBC (t_(17)_ = 3.344, *p* = 0.0038; unpaired Student’s *t*-test). (**C**) Summary graph showing the area under the curve (AUC) during phase 2 of the formalin assay in male mice treated with vehicle or CBC (t_(17)_ = 2.768, *p* = 0.0132; unpaired Student’s *t*-test). * *p* < 0.05, ** *p* < 0.01.

**Figure 4 biomedicines-12-00083-f004:**
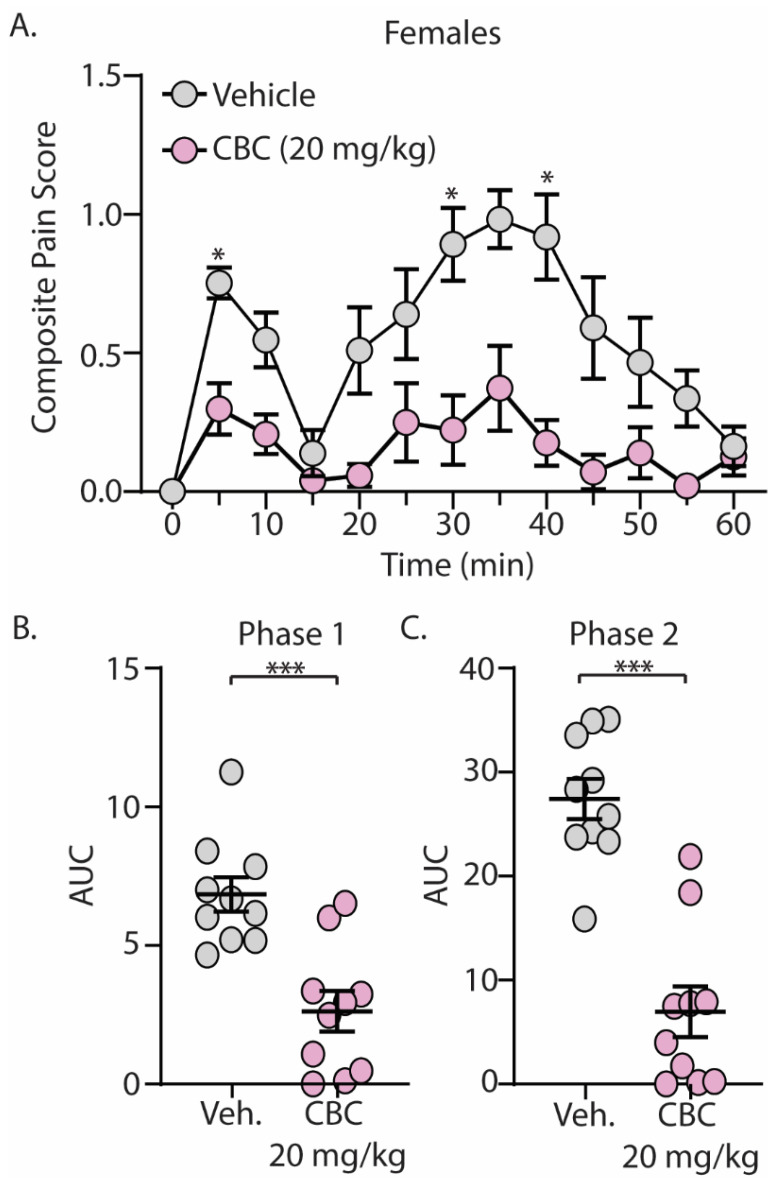
CBC produces analgesia-like effects in female mice in the formalin test. (**A**) Summary of the composite pain score from 0–60 min after formalin injection in female mice treated with vehicle or CBC (20 mg/kg, i.p.) (F_(12,216)_ = 2.607, *p* = 0.0029; two-way repeated-measures ANOVA with Bonferroni post-test). (**B**) Summary graph showing the area under the curve (AUC) during phase 1 of the formalin assay in female mice treated with vehicle or CBC (t_(18)_ = 4.416, *p* = 0.0003; unpaired Student’s *t*-test). (**C**) Summary graph showing the area under the curve (AUC) during phase 2 of the formalin assay in female mice treated with vehicle or CBC (t_(18)_ = 6.604, *p* < 0.0001; unpaired Student’s *t*-test). * *p* < 0.05, *** *p* < 0.001.

**Figure 5 biomedicines-12-00083-f005:**
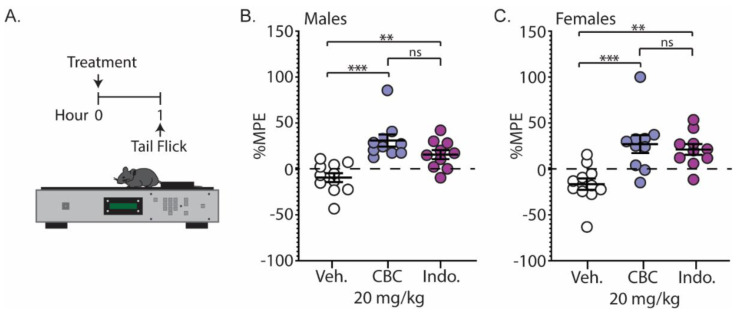
CBC produces analgesia-like effects in naïve male and female mice in the tail-flick test. (**A**) Experimental timeline for the tail-flick test. (**B**) Summary graph showing the percentage maximum possible effect (%MPE) in male mice 1 h after treatment with vehicle, CBC (20 mg/kg, i.p.) or indomethacin (10 mg/kg, i.p.) (F_(2,28)_ = 14.05, *p* < 0.0001; one-way ANOVA with Bonferroni post-test). (**C**) Summary graph as described in (**B**) in female mice (F_(2,28)_ = 10.44, *p* = 0.0004; one-way ANOVA with Bonferroni post-test). ns = not significant, ** *p* < 0.01, *** *p* < 0.001.

**Figure 6 biomedicines-12-00083-f006:**
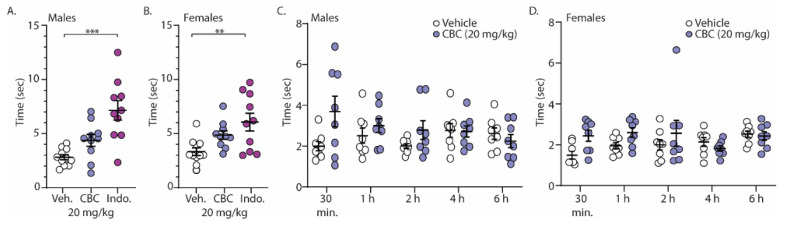
CBC effects in the acetone test in both male and female mice. (**A**) Summary graph showing the time to evoke a behavioral response following acetone exposure to the hind paw in naïve male mice 1 h following vehicle, CBC (20 mg/kg, i.p.), or indomethacin (10 mg/kg, i.p.) (F_(2,28)_ = 13.24, *p* < 0.0001; one-way ANOVA with Bonferroni post-test). (**B**) Summary graph of the same procedure outlined in (**A**) in naïve female mice (F_(2,28)_ = 6.321, *p* = 0.0054; one-way ANOVA with Bonferroni post-test). (**C**) Summary graph showing the time to evoke a behavioral response following acetone exposure to the hind paw in neuropathic male mice at varying time points (30 min, 1, 2, 4, 6 h) following vehicle or CBC (20 mg/kg, i.p.) injection (F_(4,56)_ = 2.596, *p* = 0.0459; two-way repeated-measures ANOVA with Bonferroni post-test). (**D**) Summary graph of the same procedure outlined in (**C**) in neuropathic female mice (F_(4,56)_ = 1.974, *p* = 0.1110; two-way repeated-measures ANOVA with Bonferroni post-test). ** *p* < 0.01, *** *p* < 0.001.

## Data Availability

All data is available upon request.

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
