# Peer review of "Antinociceptive Effects of Cannabichromene (CBC) in Mice: Insights from von Frey, Tail-Flick, Formalin, and Acetone Tests"

_biomedicines, 2023, doi:10.3390/biomedicines12010083_

Round 1

Reviewer 1 Report (Previous Reviewer 2)

Comments and Suggestions for Authors

Dear Authors,

thank your for your submission.

I have read again the manuscript and I have seen that the discussion has been improved but not the results. Even if I agree with the authors that report as animal studies require a long time (years),I think that these study are necessary to give results that human studies are not able to give. 

  In particular the use of cannabinoid in pain management is not well reported due to several problems, Therefore I think that animal studies are necessary to obtain basic data.

Therefore I think that in this format and with these results this manuscript can not be published, but this is my opinion. 

Author Response

We thank the reviewer for taking the time to review our manuscript. We agree, that identifying receptor targets that mediate CBC’s behavioral effects are an important next step and that these experiments are required prior to human use. Because of this, we have included this point in the Conclusion section of the Discussion. 

Reviewer 2 Report (Previous Reviewer 1)

Comments and Suggestions for Authors

nothing

Comments on the Quality of English Language

nothing

Author Response

We thank the reviewer for taking the time to review our manuscript. We feel that the revised version is much stronger based on the feedback we received during the first round of review. 

Round 2

Reviewer 1 Report (Previous Reviewer 2)

Comments and Suggestions for Authors

Dear Authors,

thank you again for your submission.

In my opinion the data are mandatory as results and not as discussion or future research.

Author Response

We thank the reviewer for taking the time to review our manuscript. We agree, that identifying receptor targets that mediate CBC’s behavioral effects are an important next step and that these experiments are required prior to human use. Because of this, we have included this point in the Conclusion section of the Discussion. Importantly, our understanding of the review process is that the reviewer, based on the hypothesis or question posed, must assess the accurate implementation of the experimental design, appropriate statistical analyses, and based on the results, the validity of interpretations. Our manuscript was carefully designed to answer a specific question within a defined scope. While we appreciate the suggestion for additional experiments, it is essential to balance the pursuit of a comprehensive study with the practical constraints of a single manuscript. Expanding the scope significantly would not only alter the trajectory of the work but may also introduce more questions, potentially changing the targeted journal. We believe that maintaining a focused approach allows for a more impactful presentation of the research without compromising the manuscript's clarity and purpose. We aim to address your concerns within future work that will seek to investigate the mechanisms of CBC action through rigorous experimental design of a multitude of receptor targets.

This manuscript is a resubmission of an earlier submission. The following is a list of the peer review reports and author responses from that submission.

Round 1

Reviewer 1 Report

Comments and Suggestions for Authors

In the paper entitled “Antinociceptive Effects of Cannabichromene (CBC) in Different Animal Models of Pain”, the authors reported the capability of CBC to reduce pain in neuropathic mice, as well as in acute, inflammatory and hot thermal pain in naïve animals.  Although lacking a real novelty because this is not the first study to demonstrate antinociceptive properties of CBC, the paper is carried out with rigor and linearity and adds new information about the range of the antinociceptive activities of CBC and it can in any case be useful to researchers interested in the subject. In my opinion, it can be published with same revisions.

Page 1 line 40: what do the authors mean when they define CBC as a “non-psychoactive” cannabinoid given that, as the authors themselves state, it has yet to be studied?

Page 2 line 67: the authors say: “The total number of mice used in this study included male (N = 58) and female (N = 59)”. However on the basis of the number of experiments carried out and the various doses of CBC and/or reference analgesic used, and what the authors themselves state in the exposition of the materials and methods, the animals used should be of a different number.

Page 3 line 91: Why was indomethacin chosen as the reference drug for neuropathic pain? In my opinion it is appropriate for formalin test experiments but not really for those on neuropathic pain

The comment on the results made in point 3.1 is not clear. The authors are talking about 

“stronger similarity in the binding profiles of the two compounds”… which ones? The authors comment further on the results obtained.

page 10 line 324: CIPN. It is the first time this acronym appears without explanation.

Discussion: the discussion is very weak. The authors argue the effects related to other cannabinoids studied in previous works but do not give sufficient hypotheses on the results obtained with CBC in the work under examination.

Comments on the Quality of English Language

Minor editing of English language required

Author Response

Many thanks for the positive comments

Page 1 line 40: what do the authors mean when they define CBC as a “non-psychoactive” cannabinoid given that, as the authors themselves state, it has yet to be studied? We apologize for any confusion, while the potential medicinal properties of CBC have largely been under-studied, nearly all studies describe CBC as a non-psychoactive cannabinoid (PMIDs 25160710, 20619971, 31368508, 37721989, 37721988).  The compound has not been reported to induce euphoria or other characteristics of psychoactive substances such as THC, while it is true that the function of CBC at the CB1 receptor is still unclear, the literature does show that if it is acting as an agonist at this receptor it does so weakly and does not induce euphoria.   

Page 2 line 67: the authors say: “The total number of mice used in this study included male (N = 58) and female (N = 59)”. However on the basis of the number of experiments carried out and the various doses of CBC and/or reference analgesic used, and what the authors themselves state in the exposition of the materials and methods, the animals used should be of a different number.  We thank the reviewer for catching this oversight, we failed to update the number in the methods after conducting some additional experiments.  This has now been corrected.

Page 3 line 91: Why was indomethacin chosen as the reference drug for neuropathic pain? In my opinion it is appropriate for formalin test experiments but not really for those on neuropathic pain  We respectfully disagree with the reviewer on this matter, other studies have shown indomethacin is effective for reducing pain-associated behavior responses in murine models of neuropathic pain and our own data presented in Figure 2 demonstrate that indomethacin reduces sensitivity to von Frey stimulus.  We have also added literature support for our use of indomethacin.

The comment on the results made in point 3.1 is not clear. The authors are talking about “stronger similarity in the binding profiles of the two compounds”… which ones? The authors comment further on the results obtained.  We apologize for this confusion, the two compounds referenced were THC and THCV (discussed in the previous sentence, lines 162-163 of the revised manuscript), however we have now added a parenthetical to avoid any potential confusion.  

page 10 line 324: CIPN. It is the first time this acronym appears without explanation.  Thank you again for catching this editing mistake on our part, the acronym has been described in the revised manuscript.

Discussion: the discussion is very weak. The authors argue the effects related to other cannabinoids studied in previous works but do not give sufficient hypotheses on the results obtained with CBC in the work under examination.  We thank the reviewer for this constructive criticism, we have added some additional text describing potential mechanisms by which CBC may reduce pain in these models.  In addition to the receptors described in our prior version (CB1, CB2, TRPA1, TRPV, NaV, and adrenergic receptors) we have added discussion of the potential role of PPARs in pain.

Reviewer 2 Report

Comments and Suggestions for Authors

Dear Authors,

I have read the manuscript and I send you my comments:

1) in results please add experimental groups showing the effect of a CB1 agonist and antagonist and CB2 agonist and antagonist after the treatment with CBC

2) PLease add an experimental group showiing the effect of a treatment with NSAIds before the CBC treatment

3) Conclusions

please change

"In the current study, we set out to evaluate the antinociceptive properties of CBC. Our results showed that CBC was effective at reducing pain in almost all behavioral models, the only exception was cold thermal pain (acetone test). However, it shows that CBC may be a potential therapeutic option for treating not only thermal pain, as some studies have shown, but also other different types of pain (neuropathic and inflammatory pain). Our results in the formalin assay also corroborate previous research on the anti-inflammatory properties of CBC. This is the first study to test the therapeutic effects of CBC in a neuropathic pain model. Future studies to identify receptors that mediate the antinociceptive effects of CBC will be beneficial in understanding the mechanism by which CBC acts."

with 

"In the current animal study, we set out to evaluate the antinociceptive properties of CBC. Our results showed that CBC was effective at reducing pain in almost all behavioral models, the only exception was cold thermal pain (acetone test). However, it shows that, in experimental animal, CBC may be a potential therapeutic option for treating not only thermal pain, as some studies have shown, but also other different types of pain (neuropathic and inflammatory pain). Our results in the formalin assay also corroborate previous research on the anti-inflammatory properties of CBC. This is the first study to test the therapeutic effects of CBC in a neuropathic pain animal model. Future studies to identify receptors that mediate the antinociceptive effects of CBC will be beneficial in understanding, in animal model, the mechanism by which CBC acts.

Comments on the Quality of English Language

none

Author Response

1) in results please add experimental groups showing the effect of a CB1 agonist and antagonist and CB2 agonist and antagonist after the treatment with CBC.  We thank the reviewer for this suggestion, and are currently working on using antagonists to examine the mechanism of action by CBC.  However, given the extensive nature, time-frame, and expense of these experiments and the likelihood that different receptors may mediate the different models tested here these studies will take months to years to finalize.  And, we feel the proposed studies are beyond the scope of the current manuscript describing the potential antinociceptive properties of CBC and the conditions for which this compound may be useful.

2) Please add an experimental group showing the effect of a treatment with NSAIDs before the CBC treatment As with the above experiment, we are also interested in the effects of CBC with the current standard of care for pain management and have also begun these studies.  But, as discussed above, the experiments conducted in this paper take months to perform.  We plan to publish CBC in combination with other pain medications in a future study.

3) Conclusions

please change this change has been made.

"In the current study, we set out to evaluate the antinociceptive properties of CBC. Our results showed that CBC was effective at reducing pain in almost all behavioral models, the only exception was cold thermal pain (acetone test). However, it shows that CBC may be a potential therapeutic option for treating not only thermal pain, as some studies have shown, but also other different types of pain (neuropathic and inflammatory pain). Our results in the formalin assay also corroborate previous research on the anti-inflammatory properties of CBC. This is the first study to test the therapeutic effects of CBC in a neuropathic pain model. Future studies to identify receptors that mediate the antinociceptive effects of CBC will be beneficial in understanding the mechanism by which CBC acts."

with 

"In the current animal study, we set out to evaluate the antinociceptive properties of CBC. Our results showed that CBC was effective at reducing pain in almost all behavioral models, the only exception was cold thermal pain (acetone test). However, it shows that, in experimental animal, CBC may be a potential therapeutic option for treating not only thermal pain, as some studies have shown, but also other different types of pain (neuropathic and inflammatory pain). Our results in the formalin assay also corroborate previous research on the anti-inflammatory properties of CBC. This is the first study to test the therapeutic effects of CBC in a neuropathic pain animal model. Future studies to identify receptors that mediate the antinociceptive effects of CBC will be beneficial in understanding, in animal model, the mechanism by which CBC acts.

Round 2

Reviewer 2 Report

Comments and Suggestions for Authors

Dear Authors,

I think that requested data are mandatory for the acceptance of this manuscript

Author Response

Dear Kesinee Preprem and Reviewer 2:

We thank you and the reviewers for the very valuable input that has significantly strengthened our manuscript.  Moreover, we acknowledge the importance of the types of additional experiments requested by R2, but feel they are well outside the scope of the present studies AND would require perhaps a year to conduct in a rigorous fashion.

Below, we provide the rationale for why these experiments were not added to our study in the hope of explaining our concerns with the proposed studies in the context of the present submission.

  • in results please add experimental groups showing the effect of a CB1 agonist and antagonist and CB2 agonist and antagonist after the treatment with CBC

We did not add these studies for three reasons. First, the results would be difficult to interpret. Typically, the antagonist or agonist is administered prior to compound treatment. This ensures that the compound, in this case CBC, is unable to bind to and/or activate the receptor prior to the administration of the test compound. In performing the experiment requested, it is possible that CBC will activate pathways mediating antinociceptive effects prior to agonist/antagonist treatment. In doing so, the agonist/antagonist treatment would therefore appear to have no effect.

Second, our study is designed to investigate the actions of CBC in different pain assays. The experiments requested address the question of CBC’s mechanism of action, thus changing the focus of the manuscript and dramatically expanding the scope of the work.

Third, the experiments are difficult to perform within a reasonable time frame. To study the mechanisms of CBC action, each agonist and antagonist would have to be administered [prior to CBC administration] for each pain assay along with the required controls (N=480 mice based on 12 groups/pain assay (4 assays) with 10 mice/group). From a practical standpoint, this is not feasible. Additionally, we would also have to include other antagonists/agonists to thoroughly investigate the mechanism of action at other CBC receptor targets (e.g., TRPA1, TRPV1, PPAR, etc.).  Parenthetically, these are just the types of experiments planned for the coming years.

2)         Please add an experimental group showing the effect of a treatment with NSAIDs before the CBC treatment

These studies were not included in the resubmission for the following two reasons. First, our investigation seeks to understand the antinociceptive properties of CBC. Treating mice with NSAIDs prior to CBC treatment would affect our ability to interpret the data. It will be unclear whether the antinociceptive effects are mediated by the NSAID, CBC, or both.

Second, we have included the NSAID, indomethacin, as a positive control for quality assurance within our experimental setup and to provide a benchmark for comparison against the effects of CBC. Studying the combination effects of NSAIDs with minor cannabinoids alters the direction of our manuscript (looking for synergy), potentially rendering it disjointed without a clear focus.

We want to point out that these directions (e.g., CBC mechanisms of action and potential effects in combination therapy) are of interest to us and we value R2’s suggestions. However, we feel that these experiments may be better served for another study in order to address them rigorously.

We thank the editor and reviewer for considering our request. Attached, please find an updated version of our manuscript that we hope provides a clearer focus of our investigation. 
